# Efficient Hardware Design and Implementation of the Voting Scheme-Based Convolution

**DOI:** 10.3390/s22082943

**Published:** 2022-04-12

**Authors:** Pedro Pereira, João Silva, António Silva, Duarte Fernandes, Rui Machado

**Affiliations:** 1Algoritmi Centre, University of Minho, 4800-058 Guimarães, Portugal; a81756@alunos.uminho.pt (P.P.); a82040@alunos.uminho.pt (J.S.); antoniosilva9116@gmail.com (A.S.); rui.machado@dtx-colab.pt (R.M.); 2Associação Laboratório Colaborativo em Transformação Digital—DTx Colab, 4800-058 Guimarães, Portugal

**Keywords:** deep learning, field-programmable gate array (FPGA), sparsity, voting convolution, 3D object detection models

## Abstract

Due to a point cloud’s sparse nature, a sparse convolution block design is necessary to deal with its particularities. Mechanisms adopted in computer vision have recently explored the advantages of data processing in more energy-efficient hardware, such as the FPGA, as a response to the need to run these algorithms on resource-constrained edge devices. However, implementing it in hardware has not been properly explored, resulting in a small number of studies aimed at analyzing the potential of sparse convolutions and their efficiency on resource-constrained hardware platforms. This article presents the design of a customizable hardware block for the voting convolution. We carried out an in-depth analysis to determine under which conditions the use of the voting scheme is justified instead of dense convolutions. The proposed hardware design achieves an energy consumption about 8.7 times lower than similar works in the literature by ignoring unnecessary arithmetic operations with null weights and leveraging data dependency. Access to data memory was also reduced to the minimum necessary, leading to improvements of around 55% in processing time. To evaluate both the performance and applicability of the proposed solution, the voting convolution was integrated into the well-known PointPillars model, where it achieves improvements between 23.05% and 80.44% without a significant effect on detection performance.

## 1. Introduction

The characteristics of the data collected from a LiDAR sensor, such as the amount of data (1.3–3 million) together with its sparse and unstructured nature [1], make the adoption of traditional convolutions, to process 3D data, a very time-consuming and inefficient task [2,3]. To try to counter this problem, novel and more computationally efficient solutions have emerged with faster mechanisms [4]. These solutions aimed at taking advantage of the data sparsity to speed up the convolution operation by reducing the number of points processed, thus decreasing the computational time and resources allocated.

A few works have been published to handle the sparsity by focusing the computational power on the relevant information from the input data [5,6,7]. One of the approaches found in the literature is known as the voting scheme-based convolution [6]. The authors proved that this convolution is mathematically equivalent to the traditional one and is able to optimize the processing of sparse data.

Given the success of Convolutional Neural Networks (CNNs) in complex tasks such as object recognition [2], variants have emerged for 3D data processing, however, the sparse and unstructured nature of the point cloud has forced the literature to increase the complexity of 3D models [1], making their implementation in edge devices unfeasible [3]. On the other hand, with the need for distributed computing in critical applications, such as autonomous vehicles [8], the availability of platforms capable of executing and accelerating CNNs is essential to enable their real-time execution. Considering the Programmable Gate Arrays (FPGAs) as a technology with the potential to deploy deep learning-based models [9], there is a need for an investigation focused on flexible solutions and their efficient design and implementation in hardware.

This article presents a hardware architecture for the voting scheme-based convolution. The proposed architecture design allows the configuration of parameters such as kernel size, padding, and stride, enabling its integration in any 3D object detection model. This is the first research exploring and proposing a design for the voting scheme-based convolution in hardware, extended with the integration of optimization mechanisms to the base operation. The implemented architecture is able to take advantage of spatially close values in the feature map and the presence of null weights. In both scenarios, it is possible to optimize the pipeline processing, either by reducing the communication with the output memory or by discarding unnecessary calculations. The Results section presents the voting convolution performance according to several metrics and the impacts of each proposed optimization mechanism are also shown. With the proposed position management of non-null output values, it is possible to fulfill the voting requirements and cascade multiple voting convolution blocks for consecutive sparse data processing.

This article is structured as follows: Section 2 presents a brief overview of sparse mechanisms and CNN accelerator frameworks. In Section 3, the architecture design of the Voting Block is presented, with a detailed description of the pipeline approach and the integrated techniques aimed at optimizing the processing under specific conditions. The results obtained in both validation and performance tests, together with the integration with the PointPillars model, are presented in Section 4. Lastly, the conclusions of the proposed architecture are summarized in Section 5.

## 2. Related Works

With the increasing integration of 3D sensors in perception systems, such as in autonomous vehicles, deep learning-based models are forced to deal with the sparsity of the data collected by these sensors. While targetting edge devices for real-time applications, energy efficiency and computation speed requirements for inference can be difficult to fulfill, and even more so when the mechanisms adopted are not the most efficient. As the convolution is the main operation of a CNN, the research community has been focused on finding better alternatives to the traditional one, still widely adopted by deep learning-based models. In the last few years, several mechanisms optimized for sparse data processing have been presented in the literature [5,7]. Sparse convolutions aim at reducing the computational cost by ignoring the input data without relevant information, thus directing the processing only to the meaningful data.

Due to the huge data flow involved in convolution layers of CNNs, the paradigm has been changing with the migration of computing from servers to platforms known as edge devices [10]. Although edge devices have fewer resources, they are closer to the scene where the action takes place, avoiding latency in communication, thus reducing susceptibility to failures. This trend has opened up space for hardware accelerators as a solution capable of satisfying the requirements of a real-time application [9]. The reduced development time and the substantial evolution of both the technology and development tools have led FPGAs to be one of the most used accelerator platforms.

A lot of work on FPGA-based CNN implementations has been introduced in the literature [11,12], taking advantage of the hardware flexibility and performance to meet real-time application requirements. VITIS AI [13], currently supported by Xilinx, has a Deep Learning Processor Unit (DPU)-based acceleration together with an extensive integration with tools and libraries optimized to implement neural networks in hardware. Similar to Core Deep Learning (CDL) [14], it supports the main CNN operations and has direct compatibility with several popular CNNs such as ResNet [15] and YOLO [16], with ready-to-use solutions. However, these accelerators do not cover all deep learning operations, in particular sparse convolutions, such as submanifold and voting.

Despite the existence of complete solutions both in the literature and market to implement custom CNNs in hardware [17], the sparsity found in 3D data prevents accelerators from efficiently processing the data and consequently achieve better performance. To address this problem, Eyeriss [18] stores the compressed activations in DRAM, saving energy by gating operations with null activations. The null activations are detected using multiple scalar Processing Elements (PEs) in a row-stationary data flow scheme. Cnvlutin [19] also stores the activation in a compressed form, however, computation cycles involving null activations are discarded to improve performance. While [18,19] are both able to drop accesses for weights on internal buffers when multiplications with null activations are detected, Cambricon.X [20] exploits sparsity by skipping operations with null weights. In [20], non-null weights define which activations are read from the input channels, however, is not able to detect and discard operations with null activations.

While hardware accelerator architectures for dense models prove to be inefficient for sparse CNN models, novel architectures have been designed to improve the performance of convolutions on sparse data [21,22]. The authors of [21] convert the spatial convolution operation to a set of matrix-vector multiplications, which in turn requires intensive memory accesses for reading the input data. To sequence the access to input data, the approach in [22] follows an element-matrix multiplication, however, the irregularity introduced by sparsity causes imbalanced loads on the allocated processing elements.

To address these challenges, we propose the first hardware architecture for the voting scheme-based convolution presented in [6]. Moreover, hardware optimization mechanisms are also integrated in the proposed architecture to take advantage of our architecture specifications and further reduce the time spent on matrix multiplications. This type of convolution, explored only by Dominic Zeng Wang and Ingmar Posner in [5,6], directs the processing only to feature map regions with relevant information, avoiding the sliding window approach over the entire data adopted by dense convolutions, thus saving the execution time of unnecessary calculations. The voting weight values are obtained by flipping the convolutional weight axis along each dimension and the voting filter only needs to be applied at each non-null location to return the same result.

Therefore, this work contributes to the current state of the art by proposing and implementing: (1) hardware architecture for the voting convolution with configurable filter size, stride, and padding parameters; (2) use of operations with stride to significantly reduce data transfer and save computation cycles; (3) use of null weights to skip computation cycles; (4) a technique to reduce the processing time through data dependency found in spatially close non-null values from the feature map.

## 3. Design and Implementation

This section presents the hardware architecture design for the voting convolution, as well as optimization techniques to improve processing time. Figure 1 illustrates the architecture of the Voting Block, highlighting the communication with the memories outside the block. A total of five memories can be distinguished, with the main emphasis on the ones used to read and store the non-null value positions in the feature maps. Unlike traditional convolutions, the voting convolution requires information regarding the position of the values to be processed, usually available in models that perform a 2D projection of the 3D data, such as PointPillars [23] and SECOND [24]. This same information is also registered together with the output data, enabling more than one Voting Block to be instantiated.

Within the block, two functional units are distinguished, the Control Unit and the Processing Unit, which are responsible for managing all the data flow and the operations to be made, respectively. As will be further detailed, the Processing Unit was designed to perform two operations simultaneously. This operation mode is supported by a selector, which allows the use of the Output Feature Map (OFM) memory second port either for enabling the double write operation or for reading data. When the Control Unit sets the second port for the read operation, the selector works as a direct wire to feed the multiplexer. The second input port of the multiplexer is connected to the Control Unit, which is in charge of evaluating, for each iteration, if the data required for processing are saved in the double FIFO. The double FIFO is used to save each iteration result and optimize the processing of input values with data dependency by reducing the communication with the output memory.

### 3.1. Processing Unit

#### 3.1.1. Four-Stage Sequential Convolution

Figure 2 presents how the voting scheme-based convolution works. For a filter size of three, nine multiplications must be performed between the input value and the filter weights. The result of each multiplication must be added to the value already stored in the corresponding output position. After all operations have been performed, the results must be written back into the output memory.

The Voting Block Processing Unit can then be represented as a set of DSPs, proportional to the filter size used, where multiplications and additions are performed simultaneously. For the example illustrated, the input is a 6 × 6 Feature Map (FM) with only one non-null value and the convolution operation is performed with stride of 1. Assuming a filter size of 3 × 3, nine Digital Signal Processors (DSPs) would be responsible for multiplying the input value by one of the voting weights and also perform the sum with the value coming from the output memory. Thereby, each one of the DSPs will have its weight and partial sum value specifically associated. However, they all share the same input value where the convolution will be applied.

#### 3.1.2. Pipeline-Based Double Computing

According to voting requirements, read and write operations to the output memory are required for each input value to be processed. One of the solutions that allows simultaneous read and write in the same memory is the Dual Port Block Ram. This memory block is composed of two ports, both able to operate as read or write. Given this feature, we propose a pipeline processing to increase performance, as shown in Figure 3, by reducing the number of clock cycles needed to complete the convolution.

The processing of two consecutively positioned non-null values in the input memory shares certain positions in the output memory. For the first convolution, the Z0–Z8 positions are read from the output memory and after all operations, and the P0–P8 values are written in the same positions. For the processing of D1, some output memory positions are shared with those read for D0 processing. Namely, the positions identified with values P1, P2, P4, P5, P7, and P8. These positions correspond simultaneously to the output of D0 processing and to the input of D1 processing.

Given this scenario, there is a possibility of leveraging the data dependency and sharing data directly between different iterations. This data sharing can be done through a FIFO, represented in Figure 1, which stores each iteration output. In the case of data dependency, part of the required data is already inside the block so there is no need to communicate with the output memory.

In the example shown in Figure 4, although six values are reused from the last convolution, only two calculations are performed simultaneously. In practice, the six calculations could be parallelized inside the Processing Unit, however, only two values can be stored in the output memory at a time, which makes it useless to parallelize more than two calculations.

In the detailed example, it is possible to obtain a reduction of three clock cycles compared to the normal voting convolution presented in Figure 3. Output values stored temporarily at the end of each pipeline are replaced each time a new input value is completely processed.

With the integration of the technique illustrated in Figure 4, in certain pipeline iterations, a maximum of two calculations are performed simultaneously. Evolving from the approach shown in Figure 2, the Voting Block Processing Unit is now composed of only two DSPs, as detailed in Figure 5. Each DSP needs to be configured to perform multiplication and addition operations, thus enabling the integration of the data reuse technique.

#### 3.1.3. Weight-Based Optimization

According to the equation presented in Figure 6, Z is the value read from the output memory at position (*x, y*), and P the value that will be written in the output memory at the same position. When the weight value (W) is null, the equation can be simplified to **P = Z**, meaning that the value stored in the output position before processing a given input value will be the same after the process finishes.

With **x** being the number of non-null values, **y** the filter size, and **z** the number of null weights in a given filter, the number of clock cycles required to complete the voting convolution can be roughly represented by Equation (Equation 1). The equation only reflects the clock cycle reduction considering the presence of null weights, however, as will be described later, other factors also influence the voting convolution performance.
(1)$$\mathrm{clock}\_\mathrm{cycles}=x*(y^{2}-z).$$

Each time a null weight is found, the multiplication between **D** and zero can be simply ignored. To reduce unnecessary operations, the block Control Unit should identify these occurrences and remove from the pipeline the iterations with a null weight value. Null weights are initially stored in the data structure *weight buffer* represented in Figure 1 so the Control Unit can manage which calculations should be cut from the pipeline. In the example illustrated in Figure 6, the presence of two null weights allows two iterations to be removed which reduces processing by two clock cycles.

#### 3.1.4. Data Quantization

In resource-limited, high-performance, and low-latency scenarios, data quantization is required to compress the model according to the memory available in the target board and to enhance power efficiency and performance in terms of inference time. In situations where the network is running on a platform with low resource limitations, the data are usually a 32-bit floating point. To enable the hardware to process the data and reduce hardware design complexity, data quantization is adopted. Targeting an integration in a model that operates with floating-point numbers, input data need to be quantized before participating in the Processing Unit internal operations. In our design, the Xilinx Floating-Point Operator IP is used to perform this quantization. The quantization level can be customized individually for the weights and the feature map values.

According to the configuration made by the user, the Voting Block controller configures the Processing Unit so the DSPs operate correctly according to the input value format. After the DSPs perform the calculations, the result is converted to floating point, before being written to memory. All conversions must be performed if the Voting Block is isolated, however, if the block is integrated into a hardware CNN, only the first one should do the input quantization. Figure 7 illustrates where the Xilinx Floating-Point Operator IP is connected within the Voting Block to perform the quantization of feature map values, weights, and output partial values. As they are three distinct values, three IPs are instantiated and connected to the inputs of the two DSPs. Counting with the dequantization of the output values from the DSPs, a total of seven Floating-Point Operator IPs are used inside the processing unit as both DSPs share the same input value D.

### 3.2. Control Unit

#### 3.2.1. Output Reference Generation

One of the defined voting requirements is the position where the non-null values are located in the feature map and their amount. No module was designed to calculate/extract the position of non-null input values (only output values). Instead, this is one of the voting requirements. As mentioned before, this information is usually available in models that operate with 2D projections of volumetric structures representative of the point cloud [23,24]. However, to build a processing chain using the Voting Block, each one must register the non-null value positions at the output. It is essential to extend the block functionalities to register each output value together with the corresponding position stored in memory. Furthermore, the number of values stored in the output memory must be counted to inform the next block of how many iterations are needed to complete the convolution. With this information, each block will have all the necessary data to perform voting scheme-based convolutions.

Figure 8 presents an example of processing two values, D0 and D1, accessed from the positions stored in the input reference memory. The scheme helps to describe the consecutive processing by a single filter and the order of values that are written to output memory along with their positions. In this particular case, D0 and D1 are spatially close, meaning that some output positions will be shared, which requires extra management to avoid replicated positions in the output reference memory.

The management of the output positions is performed using a FIFO, targeting data dependency of spatially close values. The FIFO size is proportional to the filter size, regardless of the feature map size as there is only data dependence on spatially close values. Each time a new value is written to the output memory, the control unit needs to check if its position is already stored in the FIFO. If not, its position is stored in the output reference memory and the FIFO. At the same time, the number of output non-null values is incremented. However, if the position is registered in the FIFO then it is already flagged as having a non-null output value.

#### 3.2.2. User Configuration

Although the customization level of the block allows only a few modifications to be implemented on top of the voting scheme-based convolution mechanism, the parameter values are important to correctly configure the desired convolution. Initially, the user should specify both the filter and the Input Feature Map sizes used in the convolution. According to the voting requirements, the filter must have an odd size so that a central position is used as a reference to apply the filter in a certain region.

Along with these parameters, the user must also specify both padding and stride values. Padding and stride are two common parameters in convolutions, however, only two value options are provided for each, as they cover the vast majority of the requirements found in convolutions. For all positions to be read and their values processed, it is critical to inform the Voting Block how many values there are in the input to be processed. If no value is specified, the block will set as default the total number of values in the feature map. Although this ensures that the convolution will be performed correctly, processing all feature map values removes all the efficiency of voting convolution.

Regarding the data quantization, the user should specify how many bits must be allocated for the integer and fractional parts individually for a better control over the data resolution.

## 4. Results

To validate the Voting Block, a set of tests were built, which were further divided into three major categories. The first category corresponds to the group of tests aiming to logically validate the block operation, using small feature maps. The second category intends to evaluate the block performance under multiple conditions. Finally, the Voting Block was integrated in the PointPillars model as a case study, where the sparse data processing in certain layers was analyzed. From the results, it was possible to differentiate the potential of the voting scheme-based convolution compared to traditional solutions, in real case scenarios.

### 4.1. Functional Validation

The validation performed on the Voting Block at an early stage has simple characteristics and is aimed at testing whether the block hardware implementation is operating as intended. For the first validation test, a convolution between a 151 × 151 Input Feature Map (IFM) and a 3 × 3 filter was specified. Regarding the first test, Table 1 refers to the resource consumption report from Vivado for the Zynq Z-7010 board. As a result, the utilization of both LUTs and FFs is 13.43% and 12.55%, respectively. As for the energy consumption, the total on-chip power reported was approximately 0.2 Watts.

Compared to other hardware convolution implementations, the voting convolution has a low consumption of both area and power. The approach followed in [25], for an efficient convolution implementation inspired by [26], presents a total consumption of 10,832 LUTs and the number of DSPs is proportional to the filter size multiplied by the number of allocated processing elements. The declared total power consumption is 1.739 Watts for just one convolution, being almost 8.7 times higher than the consumption required by the Voting Block. As a reference, work [26] made a comparison of energy consumption against an energy-efficient reconfigurable accelerator for deep convolutional neural networks called Eyeriss [27]. Given four distinct network configurations, the one in [26] proved to be the most energy-efficient due to its optimized Broadcast, Stay, and Migration (BSM) data flow for input, weight, and output data, respectively.

Figure 9 presents the convolution result between an image of a car and the Gaussian blur filter. Despite the voting scheme-based convolution being only suitable for sparse data processing, the final result of any convolution should always be the same as using a traditional convolution mechanism. Since an image is a dense data representation, the time consumed by the Voting Block to complete the convolution is very long and even longer than the time that a traditional convolution would take. Nonetheless, the output image indicates that the convolution was performed correctly. The output image size is also correct since the padding and stride were both specified with values zero and one, respectively.

### 4.2. Sparsity Effect

The number of non-null values in the input feature map affects the data sparsity level, which can vary from zero to one, according to Equation (Equation 2).
(2)$$sparsity\left(A\right)=1-\frac{count\_nonzero(A)}{total\_elements\_of\_(A)}.$$

While zero sparsity indicates that all the input values are non-null, a sparsity with value one means that all the values are null. To evaluate the Voting Block performance in processing sparse data, several conditions should be created for the block to be subject to different levels of sparsity.

For a 512 × 512 feature map (total of 262,144 values) with sparsity levels between 80% and 100%, different performance tests were carried out to evaluate the processing time evolution according to the sparsity level variation. From Figure 10, the higher processing time registered for the Voting Block was around 4600 microseconds while the lower one was 390 (when ignoring the test with sparsity level of one, since a 100% level of sparsity means that all values are null and no time is consumed to process the data). The orange line on the graph represents the time taken by a traditional convolution, which, as can be seen, is always the same, regardless of the sparsity level.

During the various tests, the dense convolution implemented in [25] was used as a reference. Equation (Equation 3) presents an approximation of the processing time of the traditional convolution implemented in [25] according to the size of the input feature map (considering both *IFM_channels* and *OFM_channels* equal to one). Although the authors focus on an energy-efficient solution, processing parallelism mechanisms were integrated into the module architecture and will also be addressed in comparison with the voting.
(3)$$\left[\mathrm{DenseConv}\right]\mathrm{processing}\_\mathrm{time}\left(s\right)\approx \frac{IFM\_width*IFM\_height}{clock\_rate}.$$

The intersection point between the two lines is positioned around the 89% sparsity, corresponding to a processing time of about 2600 microseconds. For sparsity levels above 89%, the Voting Block is faster than the traditional convolution. In the tests performed, a clock of 100 MHz was defined, which is equivalent to 10 nanoseconds per clock cycle. The Y-axis represents the processing time related to each measurement, specified in microseconds.

### 4.3. Concentration Metric

The dispersion level of non-null values in the input feature map together with the sparsity level has an impact on the processing time obtained by the Voting Block. While sparsity refers to the number of values that need to be processed, the dispersion level is related to the proximity of these values in the feature map. Although the sparsity level is equal for both cases presented in Table 2, the processing time obtained can be quite different. Since values spatially close in the feature map belong to the same region, data can be shared during the convolution, using the technique described in Figure 3 and Figure 4.

The impact of the proposed data reuse technique is presented in Table 2. It is noted that the improvement in processing time can be more than 30% when all the non-null values are spatially close in the input feature map. The processing time improvements in the table reflect the effect of the data reuse technique when the concentration of values is maximum.

### 4.4. Null-Weight Processing Optimization

In Section 3.1.3, it was described that the pipeline processing mechanism can be optimized by removing the instructions associated with the filter null weights. From that, one more set of performance tests were carried out to evaluate how the processing time can be optimized. For each test, a 3 × 3 filter was used, and the values of the nine weights were manipulated to verify the influence of the number of null weights on the execution time.

Figure 11 shows that null weights also have a good impact on the processing time regardless of the sparsity level of the input data. For instance, although the gray line registers higher processing times than the traditional convolution (black line), it becomes faster when the number of null weights is greater than five.

### 4.5. Strided Operation Boost

The level of customization provided to the user enables both the specification of the padding and stride. While introducing padding into the voting scheme-based convolution is a simple operation, the integration of stride into a sparse convolution is the subject of study and promotes another set of performance tests on the Voting Block. The two value options (1 and 2) that the user is able to specify for the stride parameter cover most of the scenarios and interesting performance differences can be identified from the executed tests. Table 3 presents, for each stride value and sparsity level, the measurements of both processing time and number of non-null output values generated.

The most important aspect is the difference in processing time between the operation with a stride of one and a stride of two. The presented results indicate that, when the convolution is performed using a stride of two, the processing time consumed by the Voting Block is greatly reduced. This difference can be justified by the fact that, while with a stride of one all the input feature map regions (where the non-null values are located) need to be convolved with the filter, with a stride of two, some rows and columns are “discarded”. As a result of fewer values having to be read and written to output memory, the pipeline processing can be optimized and the time required to complete the convolution is reduced.

Evolving from Equation (Equation 1), and without considering the optimization mechanism targeting spatially close values, the processing time using voting convolution can be roughly represented by Equation (Equation 4).
(4)$$\left[\mathrm{VotingConv}\right]\mathrm{processing}\_\mathrm{time}\left(s\right)\approx \frac{x*(y^{2}-z)}{stride*clock\_rate}.$$

### 4.6. PointPillars

To extend the validation of the Voting Block hardware implementation, the PointPillars model was chosen as a case study to analyze the advantages of the voting scheme-based convolution. Besides being a state-of-the-art model in 3D object detection, the model meets the requirements for the Voting Block integration. As one of the critical requirements is the position of the non-null values in the feature map, that information can be accessed through the data structure that composes the Pillar Index from the PFN stage. Furthermore, the point cloud representation in a pseudo-image ensures high levels of sparsity for 2D data processing, which is ideal for evaluating the performance of a sparse convolution.

Considering that the PFN output data have high levels of sparsity, it is also relevant to analyze the sparsity evolution across the convolutional layers of each Backbone block. Since the number of null and non-null values presented in each feature map depends on the frame being processed at the moment, a frame with the highest possible number of non-null values was purposely chosen as a reference. The characteristics of the selected frame are presented in Table 4. According to the PointPillars model specification, for each one of the 64 output channels from the PFN stage, the number of non-null values can vary between 2 k and 12 k. Assuming the worst case (12 k non-null values in each channel), 768,000 values are used in total.

Following the performance tests presented in Figure 10, sparsity levels below 89% do not favor the use of the voting convolution, therefore only the first layer of block 1, in a first iteration, is a good candidate for voting-based convolutions.

To prepare the data for the different sparsity level scenarios, several frames were selected from the Kitti dataset [28]. Figure 12 presents the frames’ representation in a black-and-white format to better distinguish the amount of null values in each one. The non-null values are represented with white pixels while the null values are represented with black ones. From each pseudo-image, it is possible to recognize the captured scene and the substantial difference in the number of values that need to be processed between the feature map with the highest Figure 12a and lowest Figure 12f sparsity level. Through the visual aspect of the images, it is possible to verify that greater sparsity levels are related to scenes with few objects and narrow roads which are present in both urban and rural areas. On the other hand, the level of data sparsity also depends on the characteristics of the 3D sensor such as range and angular resolution.

Considering the range of non-null values for each of the 64 feature maps from the PFN, several tests were carried out to evaluate the Voting Block performance when processing only one of the feature maps. The blue line in Figure 13 presents the time consumption results of the Voting Block processing. As the range of non-null output values from the PFN stage for each feature map is between 2 k and 12 k, the extreme points of the blue line are identified as best- and worst-case scenarios, respectively.

Taking into account that the size of the feature maps from the PFN is 512 × 512, the dense convolution implemented in the convolutional module always consumes the same time (2621 μs) regardless of the feature maps sparsity level. Given the possibility of integrating parallelism in the dense convolution through the use of several processing elements, the gray arrows show the level of parallelism required for the processing to be as fast as the one in the Voting Block. Assuming a clock of 100 MHz, in the best-case scenario, the Voting Block only consumes 69 μs, meaning that the dense convolution needs to allocate at least 38 PEs for the processing times to be approximately equal. Since, for the worst case, the Voting Block consumes more time, only eight PEs are required to decrease the processing time from 2621 μs to 341 μs.

Considering the layers where the sparsity level favors the use of the voting scheme-based convolution, and considering the concentration level of the non-null values in the frames presented in Figure 12, the integration in the first three layers of block 1 was carried out, assuming a 200 MHz clock on the board. In each test, the time consumed by the dense convolution and the Voting Block processing the feature maps associated with these layers was measured. Regarding the dense convolution, the parallelism was set to maximum per filter to improve performance, using all available DSPs. The results are presented in Table 5, where the worst-case scenario was assumed, with each feature map from the PFN stage containing a total of 12k non-null values to process. In addition, the hardware processing time was compared with the software version, whose runtimes were measured on a desktop with an Intel i7 CPU and a 1080ti GPU.

The processing time improvements for the Voting Block, with the two opposite cases represented in the last column, are only positive for the first two layers, since the sparsity level decreasing across the layers increases the time consumed. As a result, compared with the software version, in the third layer of block 1 no improvement is verified since the time to process is 10.9% longer. On the other hand, a big improvement can be seen in the first layer of block 1, with the Voting Block being 80.44% faster. This great improvement is achieved due to the level of sparsity being higher in the first layer, and also because it is a strided operation, which helps the voting to achieve substantially lower execution times, as referenced in Table 3.

Compared to the detections obtained in the software version, in the hybrid version, the scores are a little higher for certain situations, resulting in false-positive detections for the same score threshold value. Therefore, the score threshold was increased to remove additional incorrect detections, resulting in similar detections for both versions. The detection scores for both software and hybrid versions are detailed in Table 6, and visually complemented by Figure 14. The difference in the detection score values are related to the information loss from the quantization and dequantization processes in hardware, however, the results prove that the integration of the voting convolution does not compromise the object detection.

## 5. Conclusions

Although 3D object detection models generally adopt traditional/dense convolution operations for data processing, the sparse and unstructured nature of point clouds requires the use of optimized mechanisms for more efficient processing. This article presents the hardware implementation of a sparse convolution named voting scheme-based convolution.

The proposed configurable hardware architecture allows the use of voting in different CNN layers through customizing stride, padding, and kernel size parameters. Moreover, it is able to take advantage of the spatial proximity between non-null values in the feature map and the presence of null weights to increase processing performance up to 30%. From tests performed under different sparsity levels, the Voting Block proved to be faster than the dense convolution at processing a 512 × 512 feature map with sparsity levels greater than 89%. Increasing the sparsity of the input data translates into an almost linear reduction in processing time without the need to allocate more resources.

From the integration with the PointPillars model, the performance of the Voting Block was measured for the first three layers of the Backbone stage. The results obtained regarding the processing time proved the advantage of using the voting convolution instead of a traditional one in both the *Conv0* and *Conv1* layers, with improvements of 80.44% and 23.05%, respectively, without compromising the model detection results. While voting proved to be more efficient in time, energy efficiency was also achieved with a consumption 8.7 times lower compared to a similar convolutional module from the literature.

Future work will focus on the integration of both dense and voting scheme-based convolutions in the same hardware module. As proven by this article, both convolutions have their potential, however, they should be used under different conditions.

## Figures and Tables

**Figure 1 sensors-22-02943-f001:**
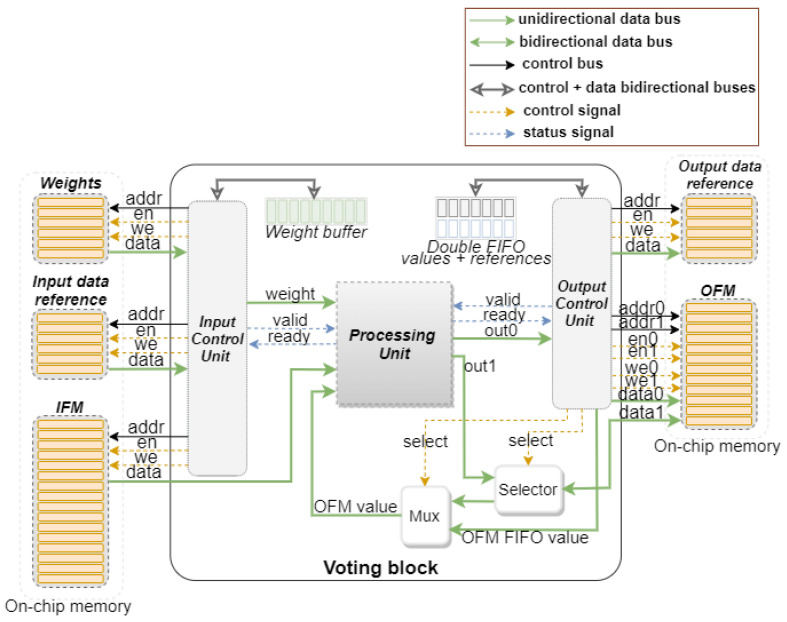
Voting Block architecture.

**Figure 2 sensors-22-02943-f002:**
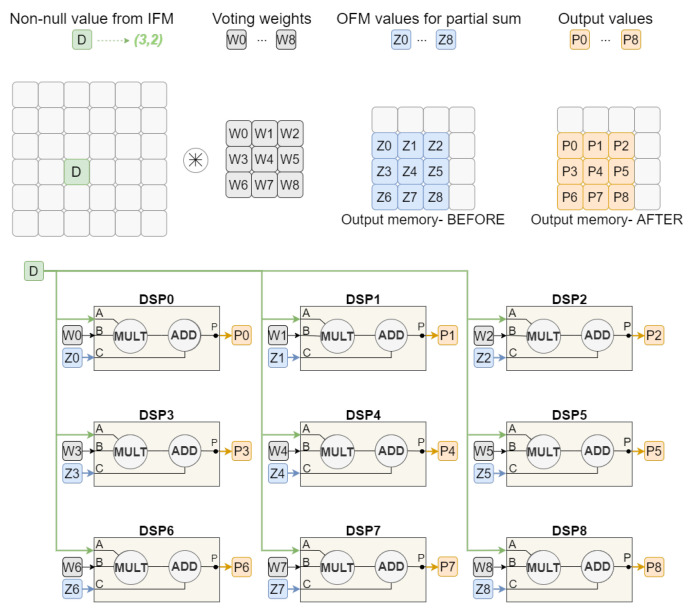
Voting convolution mechanism (first approach).

**Figure 3 sensors-22-02943-f003:**
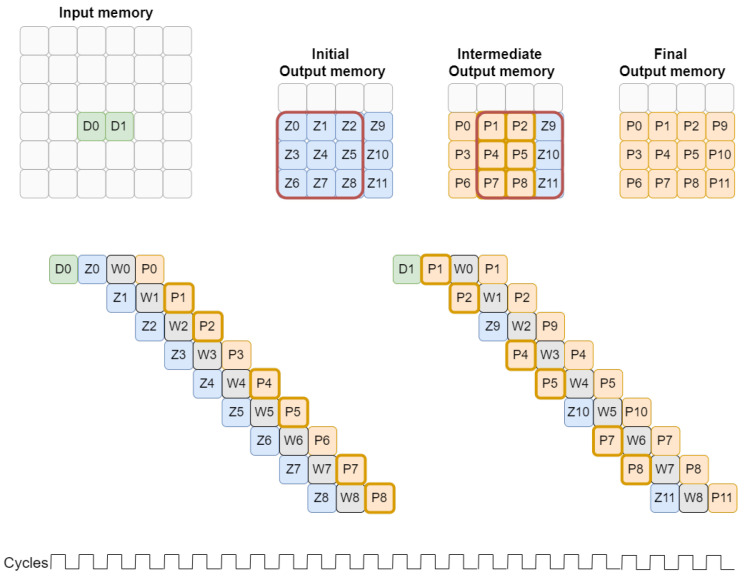
Pipeline processing with data dependency on spatially close values.

**Figure 4 sensors-22-02943-f004:**
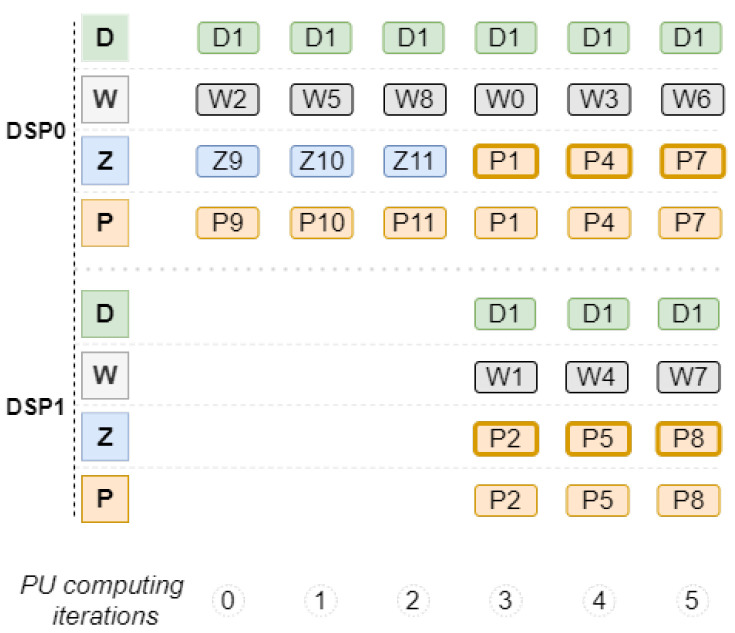
Processing Unit iterations example.

**Figure 5 sensors-22-02943-f005:**
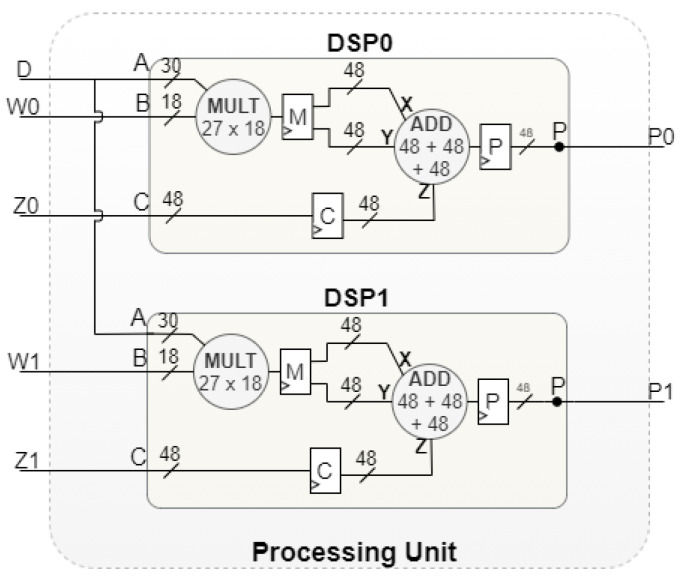
Processing Unit design architecture.

**Figure 6 sensors-22-02943-f006:**
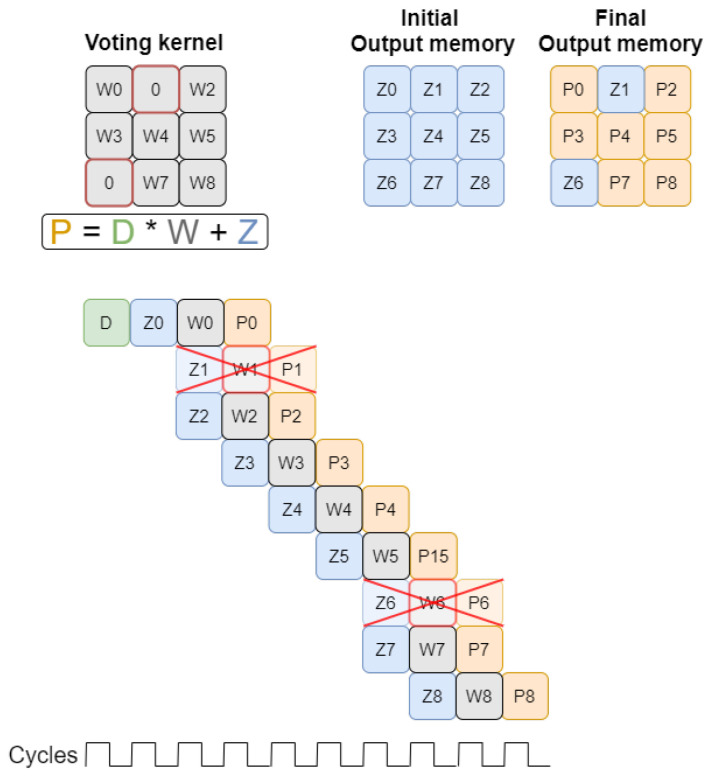
Null weight optimization.

**Figure 7 sensors-22-02943-f007:**
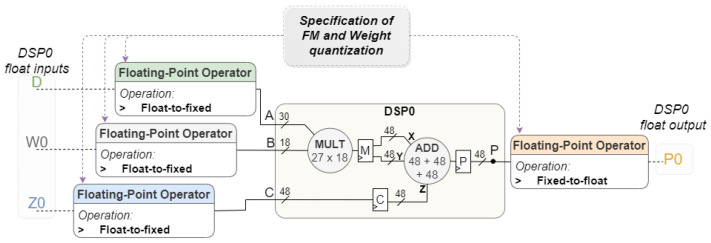
Processing Unit I/O quantization.

**Figure 8 sensors-22-02943-f008:**
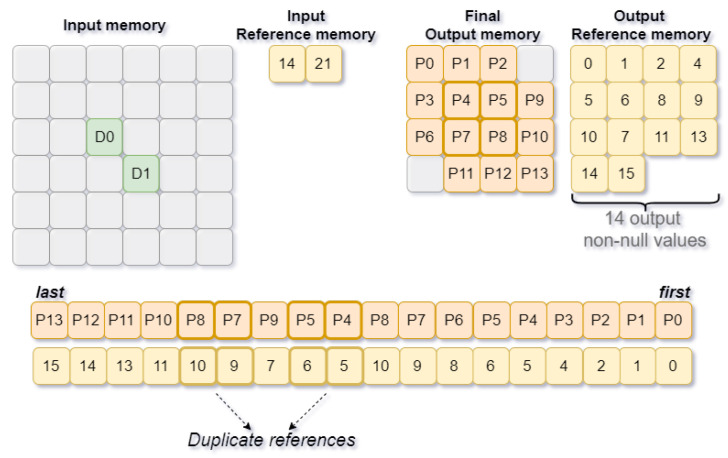
Voting Block output reference management.

**Figure 9 sensors-22-02943-f009:**
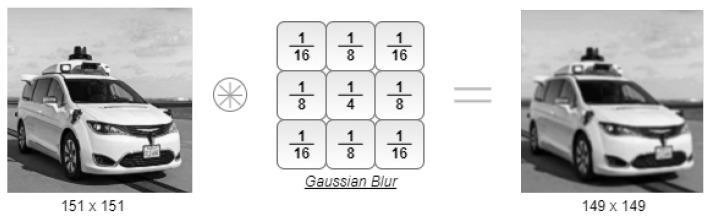
Voting convolution with a Gaussian blur filter.

**Figure 10 sensors-22-02943-f010:**
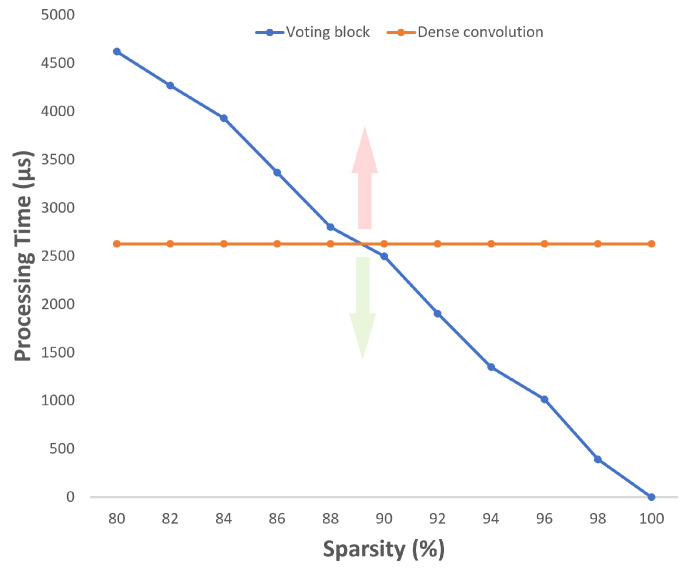
Processing time variation with increasing sparsity.

**Figure 11 sensors-22-02943-f011:**
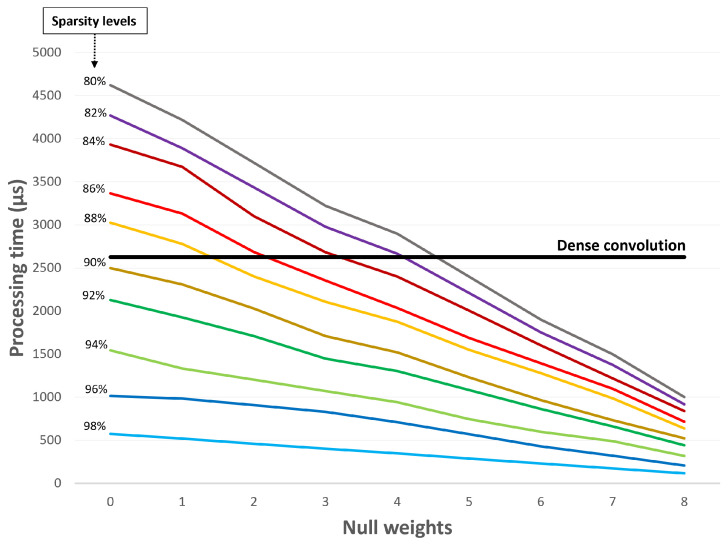
Null weight effect on processing time.

**Figure 12 sensors-22-02943-f012:**
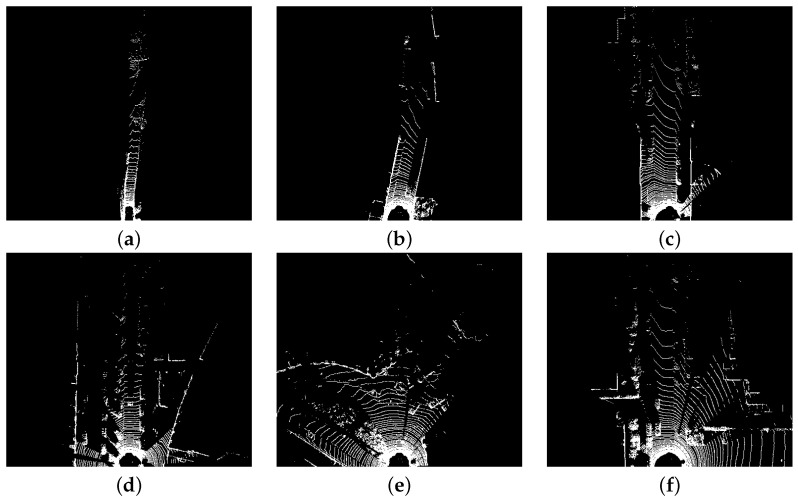
Test frames with different sparsity levels. (**a**) ≈2 k values, sparsity = 0.992. (**b**) ≈4 k values, sparsity = 0.984. (**c**) ≈6 k values, sparsity = 0.977. (**d**) ≈8 k values, sparsity = 0.969. (**e**) ≈10 k values, sparsity = 0.961. (**f**) ≈12 k values, sparsity = 0.954.

**Figure 13 sensors-22-02943-f013:**
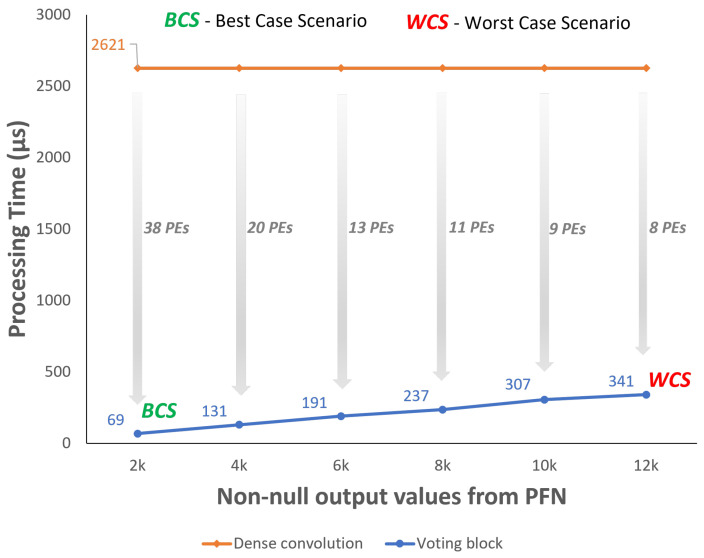
Voting processing time for the selected frames Figure 12.

**Figure 14 sensors-22-02943-f014:**
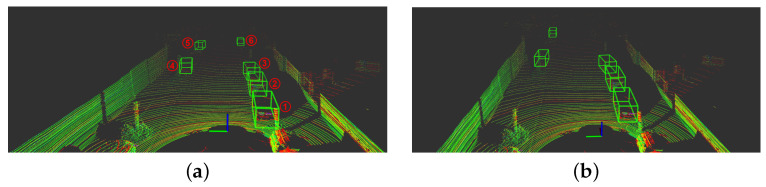
Comparison of detection between software-only and hybrid versions. (**a**) Detections with the software version (score threshold = 0.5). (**b**) Detections with the hybrid version (score threshold = 0.75).

**Table 1 sensors-22-02943-t001:** Voting Block consumption results obtained from the Vivado Report Utilization tool.

Resource	Utilization	Available	Utilization
LUT	2364	17,600	13.43
FF	4416	35,200	12.55
BRAM	1	60	1.67
DSP	2	80	2.50

**Table 2 sensors-22-02943-t002:** Value concentration effect on processing time.

Sparsity (%)	Processing Time (μs)		Improvement (%)
	Concentration (0%)	Concentration (100%)	
80	4622	3507	24.1
82	4270	3227	24.4
84	3932	2946	25.1
86	3367	2506	25.6
88	2800	2079	25.8
90	2500	1845	26.2
92	1905	1393	26.9
94	1412	1026	27.3
96	1085	773	28.8
98	392	272	30.6
100	0	0	0.0

**Table 3 sensors-22-02943-t003:** Strided voting convolution performance test.

Input Sparsity (%)	Stride = 1			Stride = 2		
	Time (μs)	Output Values	Output Sparsity (%)	Time (μs)	Output Values	Output Sparsity (%)
80	3507	53,372	79.6	1378	12,845	80.4
82	3227	48,082	81.6	1266	11,665	82.2
84	2946	42,782	83.7	1156	10,420	84.1
86	2506	37,472	85.7	989	9043	86.2
88	2079	32,181	87.7	881	7667	88.3
90	1845	26,870	89.8	714	6488	90.1
92	1393	21,580	91.8	585	5046	92.3
94	1026	16,232	93.8	441	3932	94.0
96	773	10,905	95.8	285	2490	96.2
98	272	5515	97.9	121	1245	98.1
100	0	0	100	0	0	100

**Table 4 sensors-22-02943-t004:** Sparsity level of Backbone layers.

		Non-Null Values	Null Values	Sparsity
*PFN*		768,000	16,009,216	0.95
*Block1*	*Conv0*	293,601	1,803,551	0.86
	*Conv1*	440,402	1,656,750	0.79
	*Conv2*	734,003	1,363,149	0.65
	*Conv3*	775,946	1,321,206	0.63
*Block2*		256,901	267,387	0.51
*Block3*		138,936	123,208	0.47

**Table 5 sensors-22-02943-t005:** Processing time comparison between SW version, dense convolution, and Voting Block.

	Software	Dense Convolution		Voting Block	
	Time (μs)	Time (μs)	Improvement (%)	Time (μs)	Improvement (%)
* **B1-Conv0** *	874	654	25.18	171	80.44
* **B1-Conv1** *	321	262	18.32	247	23.05
* **B1-Conv2** *	321	262	18.32	356	−10.90

**Table 6 sensors-22-02943-t006:** Detection scores of both software and hybrid versions.

Version	Frame Objects					
	1	2	3	4	5	6
*Software* (Figure 14a)	0.954	0.906	0.847	0.839	0.692	0.609
*Hybrid* (Figure 14b)	0.921	0.881	0.844	0.823	0.774	-

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
