# Peer review of "Efficient Hardware Design and Implementation of the Voting Scheme-Based Convolution"

_sensors, 2022, doi:10.3390/s22082943_

Round 1

Reviewer 1 Report

The manuscript presents a hardware design, based on an FPGA, for the voting scheme-based convolution.

In general, the manuscript is clearly described and well written; however, I recommend the manuscript to be checked by a native English speaker to improve further its readability.

Acronyms must be defined at their first appearance in the main text. For instance:

- DPU.

What do IFM, OFM and FM stand for?

Please avoid the use of possessive forms. For instance:

- iteration’s output

Although authors claim that “This is the first research exploring and proposing a design for the voting scheme-based convolution in hardware.” From this reviewer’s point of view, a comparison about used resources and performance with previous hardware implementations in the related subject will be useful to highlight the novelty and contribution of the work.

Finally, I recommend the authors to revise conclusions to highlight the novelty and contribution of their work, because at its present state, it seems more like a summary of implementation and considerations taken into account during the architecture design.

Author Response

We would like to thank you for the comments and suggestions for improvement our article. Indeed, it will help to improve and clarify our research. Bellow, the reviewer can find the performed changes to the text and reply to the questions raised by the reviewer.

Point 1: The manuscript presents a hardware design, based on an FPGA, for the voting scheme-based convolution. In general, the manuscript is clearly described and well written; however, I recommend the manuscript to be checked by a native English speaker to improve further its readability.

Response 1: Many thanks for the reviewer comment. An effort was made to improve both clarity and engagement of the manuscript.

Point 2: Acronyms must be defined at their first appearance in the main text. For instance: DPU, IFM, OFM, and FM?

Response 2: There was indeed no definition of the more technical acronyms. The definition of this acronyms has been introduced in the text as follows: Deep Learning Processor Unit (DPU), Input Feature Map (IFM), Output Feature Map (OFM), and Feature Map (FM).

Point 3: Please avoid the use of possessive forms. For instance: iteration’s output.

Response 3: We agree with the reviewer. The advice was followed and the appropriate modifications were made to the presence of possessive forms.

Point 4: Although authors claim that “This is the first research exploring and proposing a design for the voting scheme-based convolution in hardware.” From this reviewer’s point of view, a comparison about used resources and performance with previous hardware implementations in the related subject will be useful to highlight the novelty and contribution of the work.

Response 4: Comparison with similar works is in fact important to distinguish our study and to demonstrate its usefulness in the subject. However, there is a lack of works focused on configurable sparse convolutions in hardware. Typically, these deep learning methods are integrated into complex proprietary Deep Learning acceleration frameworks. Thus, information about the description of features, resources, performance and implementation with respect to the base operation is not provided. We compared our implementation with a convolutional module that implements an efficient traditional convolution. These implementations share some ideas, but the reasoning of our implementation is inherently different,  forcing us to optimize access to the memory. Aspects such as speed, energy consumption and memory utilization are analysed in detail, providing a deep comparison between these implementations.

A special emphasis on the Results chapter was performed, while the abstract and conclusion were rewritten to accommodate these insights. Besides the innovation from the hardware implementation process of the voting convolution, intensive performance comparison with the traditional convolution was detailed to highlight the novelty and contribution of the work. Moreover, a research question was raised and replied on the document. Where we analysed which set of scenarios is profitable the utilization of voting implementation to the detriment of a traditional implementation.

Point 5: Finally, I recommend the authors to revise conclusions to highlight the novelty and contribution of their work, because at its present state, it seems more like a summary of implementation and considerations taken into account during the architecture design.

Response 5: We agree with the reviewer's recommendation. Changes were made in the conclusions section to better summarize the contribution of the work, clearly highlighting the innovation achieved.

“The proposed configurable hardware architecture allows the use of voting in different CNN layers through customizing stride, padding, and kernel size parameters. Moreover, it is able to take advantage of the spatial proximity between non-null values in the feature map and the presence of null weights to increase processing performance up to 30\%.”

Reviewer 2 Report

A hardware design of a voting-based scheme of convolution is presented to deal with sparsity.

The idea to exploit the sparsity is a good idea but the design ideas have to be better presented and finally discussed as compared with similar solutions presented in the literature. Also it is necessary to compare the new ideas and technical contributions with similar ones to clearly put in evidence the value of these ideas. A comparison section with similar solutions existing in the literature it would be necessary. Also, some results should be better presented and discussed. For example in Fig.10 it is shown that the advantages of using a voting scheme is useful only for about 86% sparsity. Please, justify the useful of this result by presented real applications where a such result is useful.

Author Response

We thank you very much for the suggestions given for our manuscript. We do believe that the suggested changes have improved the overall quality of the manuscript. Bellow, the reviewer can find the performed changes to the text and reply to the questions raised by the reviewer.

Point 1: A hardware design of a voting-based scheme of convolution is presented to deal with sparsity. The idea to exploit the sparsity is a good idea but the design ideas have to be better presented and finally discussed as compared with similar solutions presented in the literature. Also, it is necessary to compare the new ideas and technical contributions with similar ones to clearly put in evidence the value of these ideas. A comparison section with similar solutions existing in the literature it would be necessary.

Response 1: Many thanks for the reviewer comments. We agree with the reviewer's notes, a detailed contextualization is relevant to position our work in the current state of the literature and also to demonstrate its relevance on the subject. To highlight the potential of voting convolution we employ a comparison against a traditional hardware convolution from the literature. The two metrics explored were power efficiency and processing time while trying to prove that sparse convolutions can actually be a good alternative to traditional ones in the presence of sparse data. Throughout the results section, the influence of each metric on the voting performance was presented through several use cases (sparsity level, stride value, number of null weights, and also the concentration of the values in the feature map). Thus, the focus was mainly on demonstrating that voting convolution can complement traditional convolution for more efficient and optimized processing under certain situations. We address the reviewer's recommendations and proceed to highlight the innovation and technical contribution of the work in the Abstract, Results, and Conclusion. As we intend to make an intensive comparison with a solution that prioritizes efficiency, we chose a convolutional module from the literature with an efficient processing dataflow. The module from the literature is able to leverage from parallelism mechanisms to increase the processing throughput which was also useful to compare with our work in terms of speed and resource consumption.

Point 2: Also, some results should be better presented and discussed. For example, in Fig.10 it is shown that the advantages of using a voting scheme is useful only for about 86% sparsity. Please, justify the useful of this result by presented real applications where a such result is useful.

Response 2: Figure 10 shows the comparison of the processing time between the traditional convolution and the voting with the variation of the sparsity level of the input data. Traditional convolution processing time (regardless of software or hardware implementation) depends on the amount of data to be processed but not on the data sparsity level. On the other hand, the voting convolution processing time depends on the sparsity level, translating into a consecutive reduction of processing time with increasing of sparsity (fewer non-null values to process). According to the measurements shown in Figure 10, the voting convolution is faster than traditional convolution for sparsity levels greater than 89%. We consider this test important to demonstrate for a particular case (the 512x512 feature map) that the traditional convolution is more suitable for certain conditions (sparsity < 89%) while the voting convolution is a better alternative for others situations (sparsity > 89%). In our research, we considered real data, i.e. point clouds provided by the Velodyne HDL-64, and we concluded after analysing its sparsity nature and by comparing the performance of the voting and traditional convolution implementation, that the voting is suitable for the first set of layers of any object detection model.

Therefore, we rewrite parts of the document to make clear to the reader that this study is relevant to define when convolution voting should be used instead of the traditional one in order to optimize processing. To better specify the difference in processing time a new equation (3) was introduced to define an approximation of execution time of the traditional/dense convolution chosen from the literature.

Reviewer 3 Report

  1. Figures are generally not easy to read, as the texts are small in size.
  2. Equation (2) instead of formula 2.
  3. * for multiplication or convolution?
  4. The title of the paper is 'efficient hardware' but there is no reported  efficiency in the conclusion. The reported performance improvement is in term of speed. 
  5. What is the significant of 100% sparsity? 
  6. Based on Figure 10, if 50% speed improvement is required, nearly 94% sparsity is required. However, the sparsity depends on the data, not the design. What is the overhead if the hardware has to be adaptive to cater for sparsity? At least not to less than 89% 

Author Response

We would like to thank you for the comments and suggestions for improvement of our article. Indeed, it will help to improve and clarify our research. Bellow, the reviewer can find the performed changes to the text and reply to the questions raised by the reviewer.

Point 1: Figures are generally not easy to read, as the texts are small in size.

Response 1: Many thanks for the reviewer comment. We realized that, in fact, the text readability of some figures could be improved. Changes have been made to some figures (Figs: 1, 2, 5, 7 and 9 in this regard.

Point 2: Equation (2) instead of formula 2

Response 2: This expression has been corrected as well as other similar occurrences of it.

Point 3: * for multiplication or convolution?

Response 3: The symbol in Figure 9 referred to a convolution between an image and the described filter. To better clarify the type of operation, the symbol was modified and complemented by the description of the figure.

Point 4: The title of the paper is 'efficient hardware' but there is no reported efficiency in the conclusion. The reported performance improvement is in term of speed.

Response 4: An effort was made in designing the Voting block hardware architecture to implement an efficient convolution. As the reviewer pointed out, efficiency is a generic work, and thus we should specify for which performance metrics our solution is in fact efficient.. We pay special attention to the speed and power efficiency of our solution. In order to better detail the efficiency achieved in the hardware implementation itself, the comparison with the hardware implementation of a traditional convolution was better highlighted in the revised version of the article, namely in terms of power consumption, in the conclusion. More efficiency specifications were mentioned in the Functional Validation subsection of the Results section.

Point 5: What is the significant of 100% sparsity?

Response 5: According to equation (2), sparsity equal to 1 (100% in percentage) implies that all feature map values are null (count_nonzero(A) = 0). On the other hand, when all feature map values are non-zero (count_nonzero(A) = total_elements_of _(A)) then the sparsity level is 0%. Although these are two possible events, they are both extreme cases. In real scenarios they rarely occur, however, they were addressed as references for the case study.

Point 6: Based on Figure 10, if 50% speed improvement is required, nearly 94% sparsity is required. However, the sparsity depends on the data, not the design. What is the overhead if the hardware has to be adaptive to cater for sparsity? At least not to less than 89%

Response 6: Many thanks in advance for the reviewer's comment since it is quite relevant given the study we carried out. In fact, sparsity is a characteristic of the input data. The voting convolution was designed to take advantage of sparsity to optimize data processing. Therefore, the performance of the voting block is dependent on the input data and weights of the model filters and not exclusively on the hardware architecture itself together with its optimization mechanisms. Moreover, as mentioned throughout the Design and Implementation section, for the block to operate correctly it is essential to fulfill the voting requirements, namely to provide information about the location of non-null values in the feature map. This information enables the voting block to process data with any sparsity level as the block architecture was not designed for a specific sparsity. However, our study has proved that voting mechanisms are suitable for the firsts layers of any object detection model designed for processing point clouds.

The voting block implements a type of sparse convolution and therefore is better suited for sparse data processing. There is no overhead difference between sparsity levels above and below 89%, however, for levels < 89% the processing time of a traditional convolution is lower compared to the voting one. We believe that future developments will be able to integrate both types of convolutions into the same hardware module in order to choose the most optimized operation for the characteristics of the input data.

Round 2

Reviewer 2 Report

The paper has been improved and I think it can be accepted with some improvements but still there are some points where it can be improved:

-the presentation could be still be improved to better highlight the technical contributions specially related to the hardware design

-the comparison section could be also improved

-please, give some real applications where the sparsity level over 89% is really necessary

Author Response

We would like to thank you again for the comments and suggestions to improve our article. Indeed, it will help to better clarify our research. Bellow, the reviewer can find the performed changes to the text and reply to the questions raised by the reviewer.

Point 1: The paper has been improved and I think it can be accepted with some improvements but still there are some points where it can be improved: The presentation could be still be improved to better highlight the technical contributions specially related to the hardware design; The comparison section could be also improved.

Response 1: Many thanks again for the reviewer comment. We compared our implementation with a convolutional module that implements an efficient traditional convolution. These implementations share some ideas, however our implementation is inherently different enabling the voting block to leverage from different scenarios including the parameters values variation (e.g. weight values and stride) to increase even more the processing throughput. Aspects such as speed and energy consumption were analyzed in detail providing a deep comparison between these implementations.

Regarding the hardware design, the architecture was designed from scratch. An effort was made from the beginning to enable the integration of optimization mechanisms that are fully detailed in the Design section along with all decisions made in the process. A note was added in the Introduction and Related Works section to highlight the contribution related to the hardware design.

Point 2: Please, give some real applications where the sparsity level over 89% is really necessary.

Response 2: In measurements from the Lidar sensor (used for example in autonomous driving applications) some data frames may contain sparsity levels above 89%. When we analyzed Kitti's dataset we were able to prove that sparsity levels can achieve more than 99% (supported by Fig. 12 in the Results section). Frames with a high level of sparsity usually occur when there are few objects (cars, buildings, trees, etc.) in the scene captured by the sensor, which can happen both in urban and rural areas. On the other hand, LiDAR characteristics such as range and angular resolution also influence the level of sparsity. In our case, we considered the Velodyne sensor with high angular resolution, however, other sensors with lower resolution can contribute to a higher level of sparsity in the measurements. The Results section was complemented with this information as it is relevant to reinforce the presence of real scenarios with sparsity levels above 89% and consequently favor the adoption of convolution voting for data processing.